# Freehand Sketch Generation from Mechanical Components

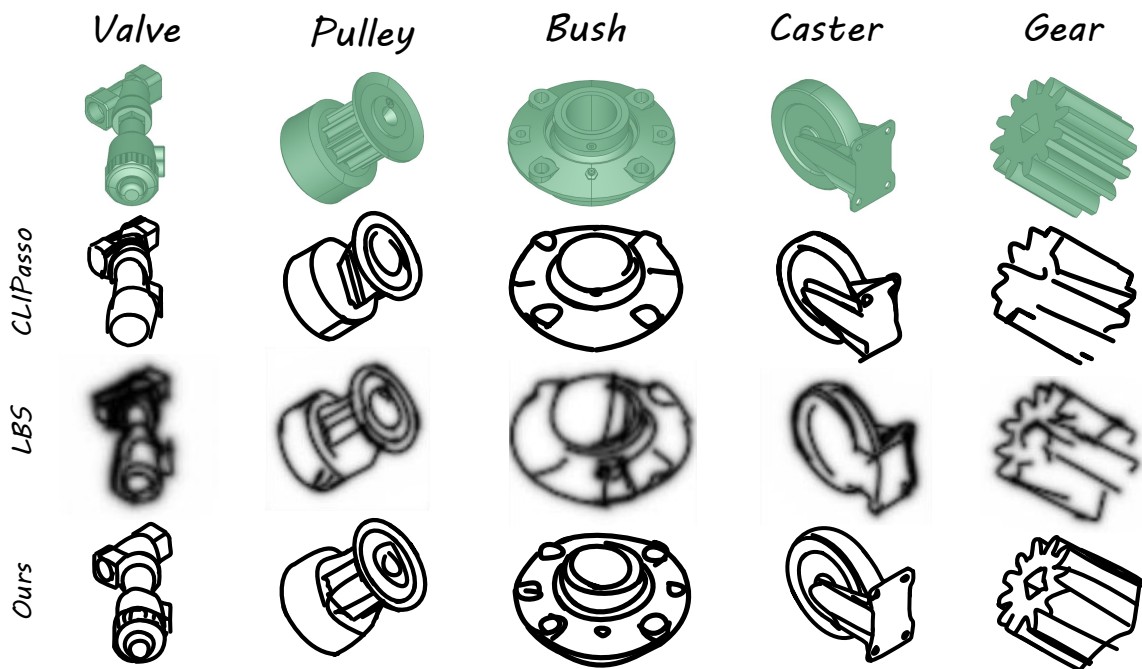

**Figure 1: Various mechanical freehand sketches generated by ours and other approaches. Our method produces sketches from mechanical components while maintaining a freehand style and their essential modeling features, e.g., grooves of the pulley, through holes on the bush, and gear teeth. (LBS[24] can't generate vector results, leading to blurriness upon enlargement.)**

## ABSTRACT

Drawing freehand sketches of mechanical components on multimedia devices for AI-based engineering modeling becomes a new trend. However, its development is being impeded because existing works cannot produce suitable sketches for data-driven research. These works either generate sketches lacking a freehand style or utilize generative models not originally designed for this task resulting in poor effectiveness. To address this issue, we design a two-stage generative framework mimicking the human sketching behavior pattern, called MSFormer, which is the first time to produce humanoid freehand sketches tailored for mechanical components. The first stage employs Open CASCADE technology to obtain multi-view contour sketches from mechanical components, filtering perturbing signals for the ensuing generation process. Meanwhile, we design a view selector to simulate viewpoint selection tasks during human sketching for picking out information-rich sketches. The second stage translates contour sketches into freehand sketches by a transformer-based generator. To retain essential modeling features as much as possible and rationalize stroke distribution, we introduce a novel edge-constraint stroke initialization. Furthermore, we utilize a CLIP vision encoder and a new loss function incorporating the *Hausdorff distance* to enhance the generalizability and robustness of the model. Extensive experiments demonstrate that our approach achieves state-of-the-art performance for generating freehand sketches in the mechanical domain.

## CCS CONCEPTS

• **Computing methodologies → Computer vision**.

## KEYWORDS

Freehand Sketch, Generative Model, Mechanical Components

**Unpublished working draft. Not for distribution.**

*ACM MM, 2024, Melbourne, Australia*
© 2024 Copyright held by the owner/author(s). Publication rights licensed to ACM.
ACM ISBN 978-x-xxxx-xxxx-x/YY/MM
https://doi.org/10.1145/nnnnnnn.nnnnnnn

## 1 INTRODUCTION

Nowadays, with the vigorous development of multimedia technology, a new mechanical modeling approach has gradually emerged, known as freehand sketch modeling [25, 26]. Different from traditional mechanical modeling paradigms, freehand sketch modeling on multimedia devices does not require users to undergo prior training with CAD tools. In the process of freehand sketch modeling for mechanical components, engineers can utilize sketches to

 

achieve tasks such as component sketch recognition, components fine-grained retrieval based on sketches, and three-dimensional reconstruction from sketches to components. Modeling in this way greatly improves modeling efficiency. However, limited by the lack of appropriate freehand sketches for these data-driven studies in the sketch community, the development of freehand sketch modeling for mechanical components is hindered. It is worth emphasizing that manual sketching and collecting mechanical sketches is a time-consuming and resource-demanding endeavor. To address the bottleneck, we propose a novel two-stage generative model to produce freehand sketches from mechanical components automatically.

To meet the requirements of information richness and accuracy for modeling, we expect that freehand sketches used for mechanical modeling maintain a style of hand-drawn while preserving essential model information as much as possible. Previous works that generate engineering sketches [15, 33, 38, 44, 52], primarily focus on perspective and geometric features of models. As a result, their sketches lack a hand-drawn style, making them unsuitable as the solution of data generation for freehand sketch modeling. Existing data-driven freehand sketch generation methods [3, 5, 11, 28, 29, 32, 43, 47, 50, 58] also fall short in this task because they require the existence and availability of relevant datasets. While CLIPasso [49] and LBS [24] can produce abstract sketches without additional datasets, as shown in Figure 1, their results for mechanical components are afflicted by issues such as losing features, line distortions, and random strokes. In contrast, we propose a mechanical vector sketch generation technique that excels in maintaining precise and abundant modeling features and a freehand style without additional sketch datasets.

Our method, the first time to generate freehand sketches for mechanical components, employs a novel two-stage architecture. It mimics the human sketching behavior pattern which commences with selecting optimal viewpoints, followed by hand-sketching. In Stage-One, we generate multi-perspective contour sketches from mechanical components via Open CASCADE, removing irrelevant information for engineering modeling which may also mislead stroke distribution in generated sketches. To select information-rich sketches, we devise a view selector to simulate the viewpoint choices made by engineers during sketching. Stage-Two translates regular contour sketches into humanoid freehand sketches by a transformer-based generator. It is trained by sketches created by a guidance sketch generator that utilizes our innovative edge-constraint initialization to retain more modeling features. Our inference process relies on trained weights to stably produce sketches defined as a set of Bézier curves. Additionally, we employ a CLIP vision encoder combining a pretrained vision transformer [8] ViT-B/32 model of CLIP [41] with an adapter [10], which utilizes a self-attention mechanism [48] to establish global relations among graph blocks, enhancing the capture of overall features. It fortifies the method's generalization capability for unseen models during training and inputs with geometric transformation (equivariance). Furthermore, our proposed new guidance loss, incorporating the *Hausdorff distance*, considers not only the spatial positions but also the boundary features and structural relationships between shapes. It improves model's ability to capture global information leading to better equivariance. Finally, we evaluate our method both quantitatively and qualitatively on the collected mechanical component

dataset, which demonstrates the superiority of our proposed framework. We also conduct ablation experiments on key modules to validate their effectiveness.

In summary, our contributions are the following:

- As far as our knowledge goes, this is the first time to produce freehand sketches tailored for mechanical components. To address this task, we imitate the human sketching behavior pattern to design a novel two-stage sketch generation framework.
- We introduce an innovative edge-constraint initialization method to optimize strokes of guidance sketches, ensuring that outcomes retain essential modeling features and rationalize stroke distribution.
- We utilize an encoder constituted by CLIP ViT-B/32 model and an adapter to improve the generalization and equivariance of the model. Furthermore, we propose a novel *Hausdorff distance*-based guidance loss to capture global features of sketches, enhancing the method's equivariance.
- Extensive quantitative and qualitative experiments demonstrate that our approach can achieve state-of-the-art performance compared to previous methods.

## 2 RELATED WORK

Due to little research on freehand sketch generation from mechanical components, there is a review of mainstream generation methods relevant to our work in the sketch community.

**Traditional Generation Method** In the early stages of sketch research, sketches from 3D models were predominantly produced via traditional edge extraction methodologies [4, 33, 35, 40, 45, 53]. Among them, Occluding contours [35] which detects the occluding boundaries between foreground objects and the background to obtain contours, is the foundation of non-photorealistic 3D computer graphics. Progressions in occluding contours [35] have catalyzed advancements in contour generation, starting with Suggestive contours [7], and continuing with Ridge-valley lines [37] and kinds of other approaches [17, 33]. A comprehensive overview [6] is available in existing contour generalizations. Similarly to the results of generating contours, Han et al. [15] present an approach to generate line drawings from 3D models based on modeling information. Building upon previous work that solely focused on outlines of models, CAD2Sketch [14] addresses the challenge of representing line solidity and transparency in results, which also incorporates certain drawing styles. However, all of these traditional approaches lack a freehand style like ours.

**Learning Based Methods** Coupled with deep learning, sketch generation approaches [3, 5, 11, 29, 38, 47, 56, 58] have been further developed. Combining the advantage of traditional edge extraction approaches for 3D models and deep learning, Neural Contours [29] employs a dual-branch structure to leverage edge maps as a substitution for sketches. SketchGen [38], SketchGraphs [44], and CurveGen and TurtleGen [52] produce engineering sketches for Computer-Aided Design. However, such approaches generate sketches that only emphasize the perspective and geometric features of models, which align more closely with regular outlines, the results do not contain a freehand style. Generative adversarial networks (GANs) [12] provide new possibilities for adding a freehand

style to sketches [11, 27, 30, 34, 51]. These approaches are based on pixel-level sketch generation, which is fundamentally different from how humans draw sketches by pens, resulting in unsuitability for freehand sketch modeling. Therefore, sketches are preferred to be treated as continuous stroke sequences. Recently, Sketch-RNN [13] based on recurrent neural networks (RNNs) [57] and variational autoencoders (VAEs) [19], reinforcement learning [9, 54, 60], diffusion models [32, 50] are explored for generating sketches. However, they perform poorly in generating mechanical sketches with a freehand style due to the lack of relevant training datasets. Following the integration of Transformer [48] architectures into the sketch generation, the sketch community has witnessed the emergence of innovative models [28, 43, 52]. CLIPasso [49] provides a powerful image to abstract sketch model based on CLIP [41] to generate vector sketches, but this method will take a long time to generate a single sketch. More critically, CLIPasso [49] initializes strokes by sampling randomly, and optimizes strokes by using an optimizer for thousands of steps rather than based on pre-trained weights, leading to numerical instability. Despite LBS [24] being an improvement over Clipasso [49], it performs unsatisfactorily in generalization capability for inputs unseen or transformed. Compared to many previous approaches, our proposed generative model can produce vector sketches based on mechanical components, persevering key modeling features and a freehand style, greatly meeting the development needs of freehand sketch modeling.

## 3 METHOD

We first elaborate on problem setting in section 3.1. Then, we introduce our sketch generation process that presents Stage-One (CSG) and Stage-Two (FSG) of MSFormer in sections 3.2 and 3.3.

### 3.1 Problem Setting

Given a mechanical component, our goal is to produce a freehand sketch. As depicted in Figure 2, it is carried out by stages: contour sketch generator and freehand sketch generator. We describe an mechanical component as $\mathcal{M} \in \Delta^3$, where $\Delta^3$ represents 3D homogeneous physical space. Each point on model corresponds to a coordinate $(x_i, y_i, z_i) \in \mathbb{R}^3$, where $\mathbb{R}$ is information dimension. Through an affine transformation, a 3D model is transformed into 2D contour sketches $C \in \Delta^2$, which consists of a series of black curves expressed by pixel coordinates $(x_i, y_i) \in \mathbb{R}^2$. In the gradual optimization process of Stage-Two, process sketches $\{\mathcal{P}_i\}_{i=1}^{K}$ are guided by guidance sketches $\{\mathcal{G}_i\}_{i=1}^{K}$, K is the number of sketches. Deriving from features of contour sketch $C$ and guidance sketches $\{\mathcal{G}_i\}_{i=1}^{K}$, our model produces an ultimate output freehand sketch $\mathcal{S}$, which is defined as a set of n two-dimensional Bézier curves $\{s_1, s_2, \ldots, s_n\}$. Each of curve strokes is composed by four control points $s_i = \{(x_1, y_1)^{(i)}, (x_2, y_2)^{(i)}, (x_3, y_3)^{(i)}, (x_4, y_4)^{(i)}\} \in \mathbb{R}^8, \forall i \in n$.

### 3.2 Stage-One: Contour Sketch Generator

Contour Sketch Generator (CSG), called Stage-One, is designed for filtering noise (colors, shadows, textures, etc.) and simulating the viewpoint selection during human sketching to obtain recognizable and informative contour sketches from mechanical components. Previous methods optimize sketches using details such as the distribution of different colors and variations in texture. However,

mechanical components typically exhibit monotonic colors and subtle texture changes. We experimentally observe that referencing this information within components not only fails to aid inference but also introduces biases in final output stroke sequences, resulting in the loss of critical features. As a result, when generating mechanical sketches, the main focus is on utilizing the contours of components to create modeling features.

Modeling engineers generally choose specific perspectives for sketching rather than random ones, such as three-view (Front/Top/Right views), isometric view (pairwise angles between all three projected principal axes are equal), etc. As shown in Figure 2 Stage-One, we can imagine placing a mechanical component within a cube and selecting centers of the six faces, midpoints of the twelve edges, and eight vertices of the cube as 26 viewpoints. Subsequently, we use PythonOCC[39], a Python wrapper for the CAD-Kernel Open-CASCADE, to infer engineering modeling information and render regular contour sketches of the model from these 26 viewpoints.

Generated contour sketches are not directly suitable for subsequent processes. By padding, we ensure all sketches are presented in appropriate proportions. Given that most mechanical components exhibit symmetry, the same sketch may be rendered from different perspectives. We utilize ImageHash technology for deduplication. Additionally, not all of generated sketches are useful and information-rich for freehand sketch modeling. For instance, some viewpoints of mechanical components may represent simple or misleading geometric shapes that are not recognizable nor effective for freehand sketch modeling. Therefore, we design a viewpoint selector based on ICNet [59], which is trained by excellent viewpoint sketches picked out by modeling experts, to simulate the viewpoint selection task engineers face during sketching, as shown in Figure 2. Through viewpoint selection, we obtained several of the most informative and representative optimal contour sketches for each mechanical component. The detailed procedure of Stage-One is outlined in Algorithm 1.

---

**Algorithm 1** Stage-One: Contour Sketch Generation

---

**Input:** Mechanical components
**Output** Contour Sketches of mechanical components

1: **procedure** CAD_TO_CONTOURS
2:  $I \leftarrow$ Read a mechanical component in STEP format
3:  Set OCC to HLR mode and enable anti-aliasing
4:  $V1 \leftarrow$ Acquire contour sketches of $I$ from the 26 built-in viewpoints in OCC
5:  $V2 \leftarrow$ Center the object in $V1$ and maintain a margin from edges of the picture
6:  $V3 \leftarrow$ Remove duplicates from sketches in $V2$ using the *ImageHash* library
7:  $O \leftarrow$ Filter the top $N$ contours with the most information from $V3$ using an image complexity estimator

---

### 3.3 Stage-Two: Freehand Sketch Generator

Stage-Two, in Figure 2, comprises the Freehand Sketch Generator (FSG), which aims to generate freehand sketches based on regular contour sketches obtained from Stage-One. To achieve this goal, we design a transformers-based [24, 31, 43] generator trained by

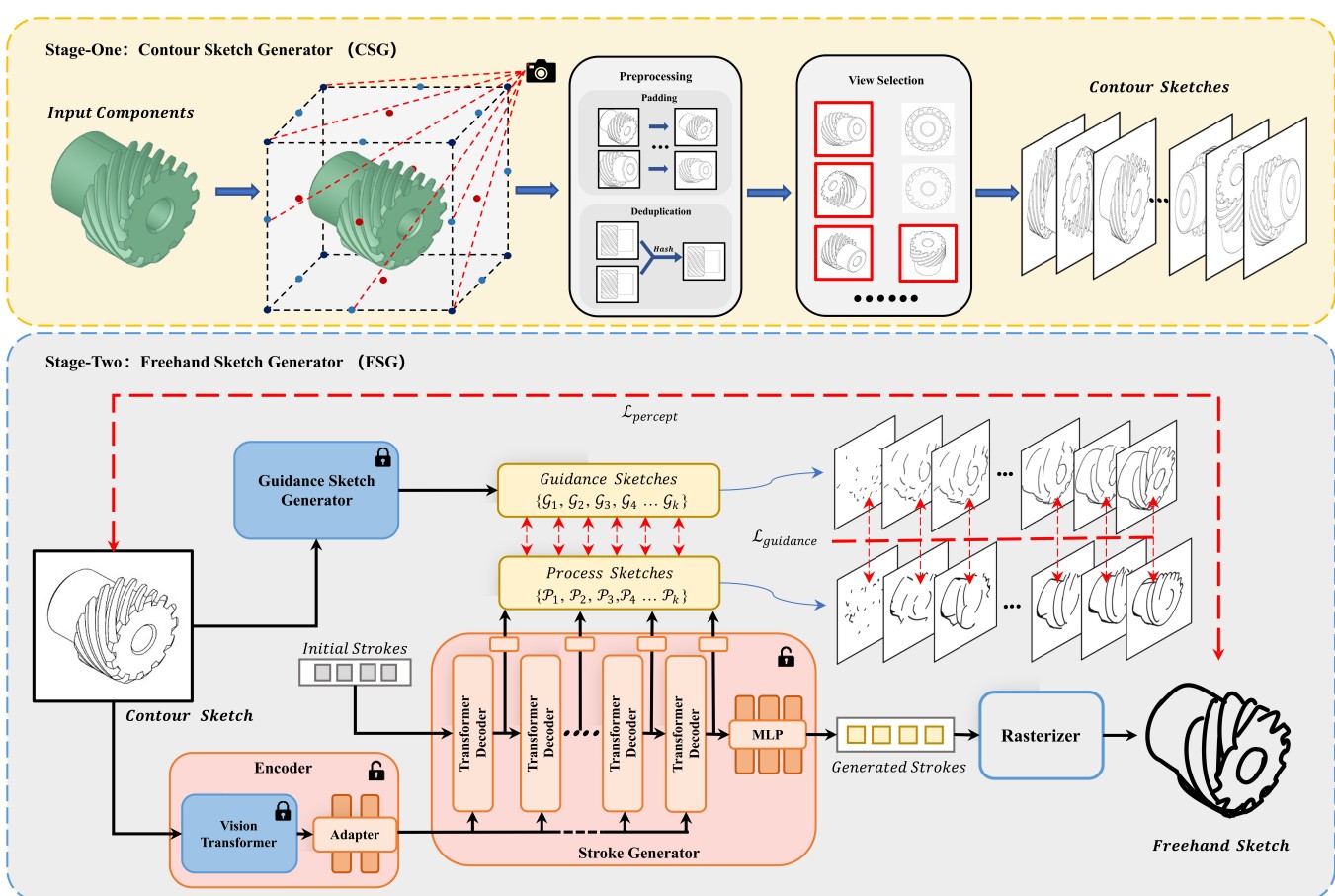

**Figure 2: An overview of our method. (1) Stage-One: we generate contour sketches based on 26 viewpoints (represented by colorful points) of a cube (grey). After that, Preprocessing and View Selection export appropriate contour sketches. (2) Stage-Two: By receiving initial strokes and features captured by our encoder from regular contour sketch, the stroke generator produces a set of strokes, which are next fed to a differentiable rasterizer to create a vector freehand sketch.**

guidance sketches, which stably generates freehand sketches with precise geometric modeling information. Our generative model does not require additional datasets for training. All training data are derived from the excellent procedural sketches produced by the guidance sketch generator.

**Generative process** As illustrated in Figure 2, freehand sketch generator consists of four components: an encoder, a stroke generator, a guidance sketch generator, and a differentiable rasterizer. Our encoder utilizes CLIP ViT-B/32[41] and an adapter to extract essential vision and semantic information from input. Although, in previous works, CLIPasso [49] performs strongly in creating abstract sketches, it initializes strokes by sampling randomly and uses an optimizer for thousands of steps to optimize sketches, resulting in a high diversity of outputs and numerical instability. To a ensure stable generation of sketches, we design a training-based stroke generator that employs improved CLIPasso[49] from the guidance sketch generator as ideal guidance. It allows us to infer high-quality sketches stably by utilizing pre-trained weights. Our stroke generator consists of eight transformer decoder layers

and two MLP decoder layers. During training, to guarantee the stroke generator learns features better, process sketches $\{\mathcal{P}_i\}_{i=1}^{K}$ (K=8 in this paper) extracted from each intermediate layer are guided by guidance sketches $\{\mathcal{P}_i\}_{i=1}^{K}$ generated at the corresponding intermediate step of the optimization process in the guidance sketch generator. In the inference phase, the stroke generator optimizes initial strokes generated from trainable parameters into a set of n Bezier curves $\{s_1, s_2, \ldots, s_n\}$. These strokes are then fed into the differentiable rasterizer $\mathcal{R}$ to produce a vector sketch $\mathcal{S} = \mathcal{R}(s_1, \ldots, s_n) = \mathcal{R}(\{(x_j, y_j)^{(1)}\}_{j=1}^{4}, \ldots, \{(x_j, y_j)^{(n)}\}_{j=1}^{4})$.

**Edge-constraint Initialization** The quality of guidance sketches plays a pivotal role in determining our outcomes' quality. Original CLIPasso[49] initializes strokes via stochastic sampling from the saliency map. It could lead to the failure to accurately capture features, as well as the aggregation of initial strokes in localized areas, resulting in generated stroke clutter. To address these issues, as shown in Figure 3, we modify the mechanism for initializing strokes in our guidance sketch generator. We segment contour sketches using SAM[20] and based on segmentation results accurately place

the initial stroke on the edges of component's features to constraint stroke locations. It ensures guidance generator not only generates precise geometric modeling information but also optimizes the distribution of strokes. Initialization comparison to original CLIPasso [49] is provided in the *Appendix*.

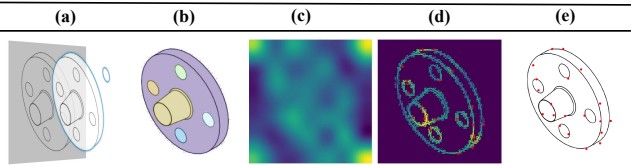

**Figure 3: Edge-constraint Initialization. (a) and (b) are results of segmenting through hole and overall segmentation of flange by SAM [20] (distinguishing features through different coloring). (c) The saliency map generated from CLIP ViT activations. (d) and (e) are initial stroke locations (in red) in final distribution map and input. It is evident that our method accurately places initial strokes at features.**

**Encoder** FSG requires an encoder to capture features. Previous works for similar tasks predominantly employ a CNN encoder that solely relies on local receptive fields to capture features, making it susceptible to local variations and resulting in poor robustness for inputs unseen or transformed. While vision transformer (ViT) uses a self-attention mechanism [48] to establish global relationships between features. It enables the model to attend to overall information in inputs, unconstrained by fixed posture or shape. Therefore, we utilize ViT-B/32 model of CLIP[41] to encode semantic understanding of visual depictions, which is trained on 400 million image-text pairs. And we combine it with an adapter that consists of two fully connected layers to fine-tune based on training data. As shown in Figure 7 and Table 1, our encoder substantially improves the robustness to unseen models during training and the equivariance.

**Loss Function** During training, we employ CLIP-based perceptual loss to quantify the resemblance between generated freehand sketch $S$ and contour sketch $C$ considering both geometric and semantic differences [41, 49]. For synthesis of a sketch that is semantically similar to the given contour sketch, the goal is to constrict the distance in the embedding space of the CLIP model represented by $CLIP(x)$, defined as:

$$\mathcal{L}_{semantic} = \phi(CLIP(C), CLIP(S)), \quad (1)$$

where $\phi$ represents the cosine proximity of the CLIP embeddings, i.e., $\phi(x, y) = 1 - \cos(x, y)$. Beyond this, the geometric similarity is measured by contrasting low-level features of output sketch and input contour, as follows:

$$\mathcal{L}_{geometric} = \sum_{i=3,4} dist(CLIP_i(C), CLIP_i(S)), \quad (2)$$

where $dist$ represents the $\mathcal{L}_2$ norm, explicitly, $dist(x, y) = \|x - y\|_2^2$, and $CLIP_i$ is the $i$-th layer CLIP encoder activation. As recommended by [49], we use layers 3 and 4 of the ResNet101 CLIP model. Finally, the perceptual loss is given by:

$$\mathcal{L}_{percept} = \mathcal{L}_{geometric} + \beta_s \mathcal{L}_{semantic}, \quad (3)$$

where $\beta_s$ is set to 0.1.

In the process of optimizing the stroke generator, a guidance loss is employed to quantify the resemblance between guidance sketches $\mathcal{G}$ and process sketches $\mathcal{P}$. Firstly, we introduce the *Jonker-Volgenant algorithm* [22] to ensure that guidance loss is invariant to arrangement of each stroke's order, which is extensively utilized in assignment problems. The mathematical expression is as follows:

$$\mathcal{L}_{JK} = \sum_{k=1}^{K} \min_{\alpha} \sum_{i=1}^{n} \mathcal{L}_1(g_k^{(i)}, p_k^{\alpha(i)}), \quad (4)$$

where $\mathcal{L}_1$ is the manhattan distance, $n$ is the number of strokes in the sketch. $p_k^{(i)}$ is the $i$-th stroke of the sketch from the $k$-th middle process layer (with a total of $K$ layers), and $g_k^{(i)}$ is the guidance stroke corresponding to $p_k^{\alpha(i)}$, $\alpha$ is an arrangement of stroke indices.

Additionally, we innovatively integrate bidirectional *Hausdorff distance* into the guidance loss, which serves as a metric quantifying the similarity between two non-empty point sets that our strokes can be considered as. It aids the model in achieving more precise matching of guidance sketch edges and maintaining structural relationships between shapes during training, thereby capturing more global features and enhancing the model's robustness to input with transformations. Experiment evaluation can be seen in section 4.5 , The specific mathematical expression is as follows:

$$\delta_H = \max\{\tilde{\delta}_H(\mathcal{G}, \mathcal{P}), \tilde{\delta}_H(\mathcal{P}, \mathcal{G})\}, \quad (5)$$

where $\mathcal{P} = \{p_1, \ldots, p_n\}$ is the process sketch from each layer and $\mathcal{G} = \{g_1, \ldots, g_n\}$ is the guidance sketch corresponding to $\mathcal{P}$. $g_i$ and $p_i$ represent the strokes that constitute corresponding sketch. Both $\mathcal{P}$ and $\mathcal{G}$ are sets containing $n$ 8-dimensional vectors. $\tilde{\delta}_H(\mathcal{G}, \mathcal{P})$ signifies the one-sided *Hausdorff distance* from set $\mathcal{G}$ to set $\mathcal{P}$:

$$\tilde{\delta}_H(\mathcal{G}, \mathcal{P}) = \max_{g \in \mathcal{G}}\{\min_{p \in \mathcal{P}}\|g - p\|\}, \quad (6)$$

where $\|\cdot\|$ is the Euclidean distance. Similarly, $\tilde{\delta}_H(\mathcal{P}, \mathcal{G})$ represents the unidirectional *Hausdorff distance* from set $\mathcal{P}$ to set $\mathcal{G}$:

$$\tilde{\delta}_H(\mathcal{P}, \mathcal{G}) = \max_{p \in \mathcal{P}}\{\min_{g \in \mathcal{G}}\|p - g\|\}. \quad (7)$$

The guidance loss is as follows:

$$\mathcal{L}_{guidance} = \mathcal{L}_{JK} + \beta_h \delta_H, \quad (8)$$

where $\beta_h$ is set to 0.8.

Our final loss function is as follows:

$$\mathcal{L}_{toatl} = \mathcal{L}_{percept} + \mathcal{L}_{guidance} \quad (9)$$

# 4 EXPERIMENTS

## 4.1 Experimental Setup

**Dataset** We collect mechanical components in STEP format from TraceParts[1] databases, encompassing various categories. On the collected dataset, we employ hashing techniques for deduplication ensuring the uniqueness of models. Additionally, we remove models with poor quality, which are excessively simplistic or intricate, as well as exceptionally rare instances. Following this, we classify these models based on ICS [2] into 24 main categories. Ultimately, we obtain a clean dataset consisting of 926 models for experiments.

**Implementation Details** All experiments are conducted on the Ubuntu 20.04 operating system. Our hardware specifications include an Intel Xeon Gold 6326 CPU, 32GB RAM, and an NVIDIA GeForce RTX 4090. The batch size is set to 32. Contour sketches from Stage-one are processed to a size of 224 × 224 pixels. Detailed information about experiments is provided in the *Appendix*.

## 4.2 Qualitative Evaluation

Due to the absence of research on the same task, we intend to compare our approach from two perspectives, which involve approaches designed for generating engineering sketches and existing state-of-the-art freehand sketches generative methods.

**Sketches of mechanical components** In Figure 4, we contrast our method with Han et al. [15] and Manda et al. [33], using our collected components as inputs. Han et al.[15] use PythonOCC[39] to produce view drawings, while Manda et al.[33] create sketches through image-based edge extraction techniques. Although their results preserve plentiful engineering features, it is apparent that their outcomes resembling extracted outlines from models lack the style of freehand, which limits applicability in freehand sketch modeling. In contrast, our approach almost retains essential information of mechanical components equivalent to their results, such as through holes, gear tooth, slots, and overall recognizable features, while our results also demonstrate an excellent freehand style.

| Input | Han et al. | Manda et al. | Ours |
|---|---|---|---|

**Figure 4: Comparison to other methods for generating sketches of mechanical components.**

**Sketch with a freehand style** We compare our method with excellent freehand sketch generative methods like CLIPasso [49] and LBS[24]. Moreover, we present a contrast with DALL-E[42], which is a mainstream large-model-based image generation approach. As shown in Figure 5, all results are produced by 25 strokes using our collected dataset. In the first example, CLIPasso's [49] result exhibits significant disorganized strokes, and LBS [24] almost completely covers the handle of valve with numerous strokes, leading to inaccurate representation of features. In the second and third examples, results by CLIPasso [49] lose key features, such as the gear hole and the pulley grooves. For LBS[24], unexpected stroke connections appear between modeling features and its stroke distribution is chaotic. In contrast, our strokes accurately and clearly are distributed over the features of components. These differences are attributed to the fact that CLIPasso[49] initializes strokes via sampling randomly from the saliency map resulting in features that

may not always be captured. Although LBS [24] modifies initialization of CLIPasso[49], it initializes strokes still relying on saliency maps influenced by noise information like monotonous colors and textures in mechanical components. Our method addresses this issue by introducing a novel edge-constraint initialization, which accurately places initial strokes on feature edges. Additionally, as LBS reported that its transformer-based model uses a CNN encoder. So its robustness comparison to our method will be similar to the results in Figure 7. In contrast to DALL-E[42], we employ inputs consistent with previous experiments coupled with the prompt ("Create a pure white background abstract freehand sketch of input in 25 strokes") as the final inputs. It is evident that the large-model-based sketch generation method is still inadequate for our task.

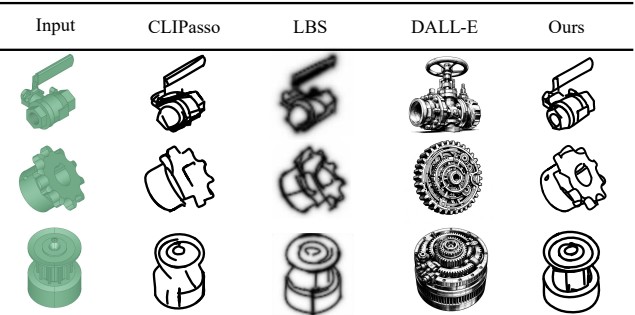

| Input | CLIPasso | LBS | DALL-E | Ours |
|---|---|---|---|---|

**Figure 5: Comparison to other state-of-the-art method for generating sketches with a freehand style.**

## 4.3 Quantitative Evaluation

**Metrics Evaluation** We rasterize vector sketches into images and utilize evaluation metrics for image generation to assess the quality of generated sketches. FID (Fréchet Inception Distance) [16] quantifies the dissimilarity between generated sketches and standard data by evaluating the mean and variance of sketch features, which are extracted from Inception-V3[46] pre-trained on ImageNet[21]. GS (Geometry Score) [18] is used to contrast the geometric information of data manifold between generated sketches and standard ones. Additionally, we apply the improved precision and recall [23] as supplementary metrics following other generative works [36]. In this experiment, we employ model outlines processed by PythonOCC[39] as standard data, which encapsulate the most comprehensive engineering information. The lower FID and GS scores and higher Prec and Rec scores indicate a greater degree of consistency in preserving modeling features between the generated sketches and the standard data. As shown in Table 1, we classify generated sketches into three levels based on the number of strokes ($NoS$): Simple ($16 \leq NoS < 24$ strokes), Moderate ($24 \leq NoS < 32$ strokes), and Complex ($32 \leq NoS < 40$ strokes). The first part of Table 1 showcases comparisons between our approach and other competitors, revealing superior FID, GS, Precision, and Recall scores across all three complexity levels. Consistent with the conclusions of qualitative evaluation, our approach retains more precise modeling features while generating freehand sketches. Additional metrics evaluation (standard data employ human-drawn sketches) is provided in the *Appendix*.

Table 1: Quantitative comparison results by metrics. "-T" means test by transformed inputs. "-U" means test by unseen inputs.

| Method | Simple | | | | Moderate | | | | Complex | | | |
|---|---|---|---|---|---|---|---|---|---|---|---|---|
| | FID↓ | GS↓ | Prec↑ | Rec↑ | FID↓ | GS↓ | Prec↑ | Rec↑ | FID↓ | GS↓ | Prec↑ | Rec↑ |
| CLIPasso [49] | 10.28 | 5.70 | 0.44 | 0.79 | 12.03 | 7.40 | 0.35 | 0.72 | 13.43 | 9.91 | 0.30 | 0.69 |
| LBS [24] | 9.46 | 5.29 | 0.45 | 0.81 | 11.57 | 7.03 | 0.32 | 0.71 | 12.71 | 8.78 | 0.31 | 0.66 |
| Ours | **6.80** | **3.37** | **0.53** | **0.87** | **7.07** | **3.96** | **0.47** | **0.83** | **7.27** | **4.52** | **0.42** | **0.81** |
| Ours(VIT-B/32+adapter) -T | 7.01 | 3.98 | 0.48 | 0.83 | 7.25 | 6.08 | 0.38 | 0.72 | 7.42 | 6.51 | 0.32 | 0.70 |
| Ours(CNN) -T | 17.46 | 28.10 | 0.18 | 0.56 | 19.44 | 63.14 | 0.13 | 0.37 | 25.13 | 79.44 | 0.11 | 0.25 |
| Ours(VIT-B/32+adapter) -U | 8.60 | 4.10 | 0.44 | 0.81 | 10.68 | 6.20 | 0.33 | 0.68 | 13.44 | 7.24 | 0.31 | 0.63 |
| Ours(CNN) -U | 18.85 | 30.33 | 0.19 | 0.51 | 20.78 | 70.66 | 0.11 | 0.40 | 27.54 | 87.54 | 0.10 | 0.20 |

**User Study** We randomly select 592 mechanical components from 15 main categories in collected dataset as the test dataset utilized in user study. We compare the results produced by Han et al.[15], Manda et al.[33], CLIPasso[49], LBS[24] and our method (the last three methods create sketches in 25 strokes). We invite 47 mechanical modeling researchers and ask them to score sketches based on two aspects: engineering information and the freehand style. Scores range from 0 to 5, with higher scores indicating better performance in creating features and possessing a hand-drawn style. Finally, we compute average scores for all components in each method. As shown in Table 2, the result of user study indicates that our method achieves the highest style score and overall score. These reveal our results have a human-prefer freehand style and a better comprehensive performance in balancing information with style.

Table 2: User study results. "Information" is the engineering information content score , "Style" denotes the score of freehand style, and "Overall" is the average of these two scores.

| Method | Information↑ | Style↑ | Overall↑ |
|---|---|---|---|
| Han et al.[15] | **4.20** | 0.84 | 2.52 |
| Manda et al. [33] | 4.04 | 1.21 | 2.63 |
| CLIPasso [49] | 2.71 | 3.81 | 3.26 |
| LBS [24] | 2.94 | 3.76 | 3.35 |
| **Ours** | 3.80 | **3.84** | **3.82** |

## 4.4 Performance of the Model

Different from traditional sketch generation methods, our generative model does not require additional sketch datasets. All training sketches are produced from our guidance sketch generator, which is optimized via CLIP[41], a model pre-trained on four billion text-image pairs, producing high-quality guidance sketches. Benefiting from the guidance sketch generation process not being limited to specific categories, our method demonstrates robustness across a wide variety of mechanical components. In Figures 1 and 6, we showcase excellent generation results for various mechanical components. More qualitative results are provided in *Appendix*.

Previous works like [24] predominantly employ a CNN encoder that uses fixed-size convolution kernels and pooling layers to extract local features. It leads to the neglect of global information,

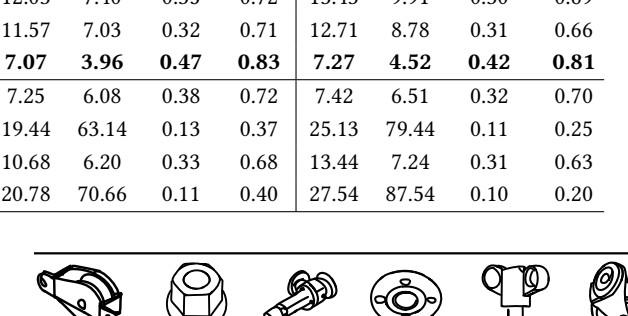

Figure 6: Robust performance across abundant categories.

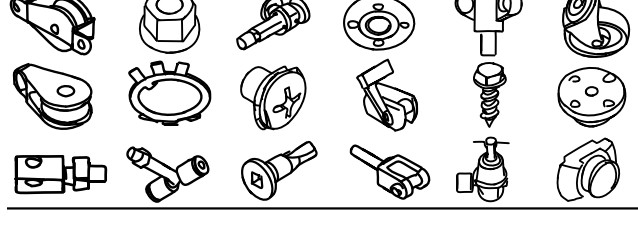

Figure 7: Comparison to our method with different encoders.

resulting in poor robustness. To address this issue, we utilize a CLIP ViT-B/32 combined with an adapter as our encoder. Qualitative and quantitative comparative experiments are designed to demonstrate the efficacy of our encoder. In the first row of Figure 7, we employ models which are similar-category, but unseen in training as test inputs. Compared to the method using a CNN encoder (ResNeXt18 [55] is used in this experiment), which only produces chaotic and shapeless strokes, the method with our encoder creates sketches with recognizable overall contours and essential features. In the second row, we apply contour sketches seen in training as inputs, each of which is transformed to the right and downward by 5 pixels and rotated counterclockwise by 10°. It can be observed that the method with our encoder still accurately infers component sketches, whereas the one using a CNN encoder fails to generate recognizable features. The quantitative comparison results are presented in the second part of Table 1. Consistent with our expectations, the method with our encoder performs better in terms of evaluation metrics. It showcases that our encoder fortifies the

**Table 3: Ablation Study with metrics evaluation. S-O: Stage-One, E-I: Guidance sketches generated by edge-constraint initialization , L-H: Training using $\mathcal{L}_{Hausdorff}$. "-T" means test by transformed inputs.**

| Model | S-O | I-O | L-H | Simple | | | | Moderate | | | | Complex | | | |
|---|---|---|---|---|---|---|---|---|---|---|---|---|---|---|---|
| | | | | FID↓ | GS↓ | Prec↑ | Rec↑ | FID↓ | GS↓ | Prec↑ | Rec↑ | FID↓ | GS↓ | Prec↑ | Rec↑ |
| Ours | | ✓ | ✓ | 9.01 | 4.73 | 0.45 | 0.81 | 10.57 | 6.79 | 0.39 | 0.75 | 11.11 | 7.20 | 0.31 | 0.68 |
| Ours | ✓ | | ✓ | 7.69 | 4.38 | 0.47 | 0.82 | 8.28 | 5.08 | 0.40 | 0.78 | 8.62 | 6.43 | 0.33 | 0.70 |
| **Ours** | ✓ | ✓ | ✓ | **6.80** | **3.37** | **0.53** | **0.87** | **7.07** | **3.96** | **0.47** | **0.83** | **7.27** | **4.52** | **0.42** | **0.81** |
| Ours -T | ✓ | ✓ | | 9.42 | 5.38 | 0.40 | 0.74 | 10.23 | 7.34 | 0.32 | 0.65 | 11.04 | 8.77 | 0.21 | 0.63 |
| Ours -T | ✓ | ✓ | ✓ | 7.01 | 3.98 | 0.48 | 0.83 | 7.25 | 5.28 | 0.38 | 0.72 | 7.42 | 6.51 | 0.32 | 0.70 |

encoding robustness for unseen and transformed inputs, enhancing the generalization and equivariance of the model.

Abstraction is an important characteristic of freehand sketches. Our method effectively achieves it by individually training the stroke generator on different levels of abstraction sketches datasets. As shown in Figure8, we respectively present the generated sketches from an input gear component using 35, 30, 25, and 20 strokes. As the number of strokes decreases, the abstraction level of gear sketches increases. Our method constrains strokes to create the essence of the gear. Iconic characteristics of a gear such as the general contour, teeth, and tooth spaces can be maintained, even though some minor details like through-holes may be removed.

**Figure 8: Different levels of abstraction generated by ours. Left to right: gear model and results in 35, 30, 25, 20 strokes.**

## 4.5 Ablation Study

**Stage-One** As shown in Figure 9, the results of the method lacking Stage-One are susceptible to issues such as producing unstructured features and line distortions in qualitative ablation experiment. Excellent metric scores in Table 3 demonstrate our complete framework can create richer and more accurate modeling information. This improvement is attributed to Stage-One, which filters out noise information such as color, texture, and shadows, mitigating their interference with the generation process.

**Edge-constraint Initialization** In order to verify whether edge-constraint initialization can make precise geometric modeling features, we remove the optimized mechanism in the initial process. Comparison in Figure 9 clearly demonstrates that sketches generated with edge-constraint initialization(E-I) exhibit better performance in details generation and more reasonable stroke distribution. These benefit from E-I ensuring that initial strokes are accurately distributed on the edges of model features. Similarly, we utilized quantitative metrics to measure the generation performance. As shown in Table 3, sketches generated after initialization optimization achieve improvements in metrics such as FID, GS, and so on.

**Hausdorff distance Loss** *Hausdorff distance* is a metric used to measure the distance between two shapes, considering not only the spatial positions but also the structural relationships between shapes. By learning shape invariance and semantic features, the model can more accurately match shapes with different transformations and morphologies, aiding in the model's equivariance. The ablation experimental result is depicted in Table 3. It is evident that all the quantitative metrics for our method training with Hausdorff distance become better on the transformed test dataset.

**Figure 9: Ablation study. E-I: Edge-constraint initialization, "Ours" are the results produced by our complete framework.**

## 5 CONCLUSION AND FUTURE WORK

This paper proposes a novel two-stage framework, which is the first time to generate freehand sketches for mechanical components. We mimic the human sketching behavior pattern that produces optimal-view contour sketches in Stage-One and then translate them into freehand sketches in Stage-Two. To retain abundant and precise modeling features, we introduce an innovative edge-constraint initialization. Additionally, we utilize a CLIP vision encoder and propose a *Hausdorff distance*-based guidance loss to improve the robustness of the model. Our approach aims to promote research on data-driven algorithms in the freehand sketch domain. Extensive experiments demonstrate that our approach performs superiorly compared to state-of-the-art methods.

Through experiments, we discover that we would better utilize a comprehensive model rather than direct inference to obtain desirable outcomes for unseen models with significant geometric differences. In future work, we will explore methods to address this issue, further enhancing the model's generalizability.

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
