# OpenReview forum: "Freehand Sketch Generation from Mechanical Components"
_acmmm.org/ACMMM/2024/Conference — MM2024 Poster_

### Official Review · Reviewer_h8pX · 2024-05-24

**Rating:** 4
**Confidence:** 2

**Summary:**

Motivated by the fact that existing methods cannot effectively generate suitable freehand sketches of mechanical components for data-driven research, this paper proposes a two-stage generative framework called MSFormer. It mimics human sketching behavior to produce humanoid freehand mechanical sketches. The first stage obtains multi-view contour sketches and filters noise. The second stage translates contours to freehand sketches using a transformer-based generator with novel techniques like edge-constraint stroke initialization and loss functions. Experiments show MSFormer achieves state-of-the-art performance for this task.

**Strengths:**

The paper is well-written and the figures are visually-pleasing.

The results are interesting and this is the my first time seeing deep learning-based freehand sketch generation of mechanical components (although it is a niche area).

The evaluation is thorough.

**Limitations:**

My major concern is the motivation of this work: why does one want to generate freehand sketches of mechanical components? "Freehand sketches" is interesting as an artistic style but if one would like to achieve high information richness and accuracy, "freehand" style is not a must. So why is this method useful?

**Suitability:**

2

---

### Official Review · Reviewer_BvKa · 2024-05-25

**Rating:** 2
**Confidence:** 3

**Summary:**

The paper introduces a new two-stage method for freehand sketch generation from mechanical components. The first stage was designed to harvest multi-view contour sketches from mechanical components while the second stage was developed to translate contour sketches into freehand sketches by a transformer-based generator. Experiments on the dataset collected from TraceParts demonstrate the effectiveness of the proposed method.

**Strengths:**

Overall, the manuscript is well-organized and easy to follow.
As claimed by the authors, this work may be a pioneering attempt toward mechanical component sketch generation.
A novel edge-constraint initialization scheme has been introduced to optimize strokes of guidance sketches.
In addition, a new guidance loss has also been developed to capture global features of sketches.

**Limitations:**

I just wonder about the application scope of the proposed method. The authors claimed in their manuscript that engineers can utilize sketches to achieve tasks such as component sketch recognition, components fine-grained retrieval based on sketches, and three-dimensional reconstruction from sketches to components. However, it seems there is no experiment to demonstrate the effectiveness of the generated sketches using the proposed method in these tasks compared to those with peer methods. For instance, compared to those sketches obtained with existing methods, what is the advantage of the sketches generated with the proposed method in facilitating the aforementioned downstream tasks?

How is the performance of the proposed method compared to the following work?
Vinker, Yael, et al. "Clipascene: Scene sketching with different types and levels of abstraction." Proceedings of the IEEE/CVF International Conference on Computer Vision. 2023.

The authors claimed that LBS [24] can’t generate vector results, leading to blurriness upon enlargement. What if using a vector graphic synthesis tool, for instance, the following one, to convert the strokes into vector graphics? Is this a fundamental limitation of LBS compared to the proposed one?
Tang, Zecheng, et al. "StrokeNUWA: Tokenizing Strokes for Vector Graphic Synthesis." arXiv preprint arXiv:2401.17093 (2024).

**Suitability:**

2

---

### Official Review · Reviewer_xoSy · 2024-06-01

**Rating:** 5
**Confidence:** 4

**Summary:**

The author designed a two-stage generative framework mimicking the human sketching behavior pattern, called MSFormer, which is the first time to produce humanoid freehand sketches tailored for mechanical components.

**Strengths:**

Impressive work! Good written skills, image drawing, and innovative ideas.
Good luck to you!

**Suitability:**

3

---

### Official Review · Reviewer_uuoC · 2024-06-02

**Rating:** 4
**Confidence:** 2

**Summary:**

This paper proposed a two-stage generative framework to create freehand sketches of mechanical components. More specifically, the first stage is to extract contour sketches by imitating human sketching behavior and the second stage is to convert the obtained contour sketches into freehand sketches using a transformer-based generator. The extensive experiments demonstrate that the proposed method can generate high-quality freehand sketches and achieve superior performances to existing state-of-the-art methods.

**Strengths:**

1. This paper appears to be the first to explore the generation of freehand sketches for mechanical components.  A two-stage framework is creatively proposed first to extract multiview contour sketches and then transform them into freehand sketches during the generation process.
2. The frozen vision encoder followed by trainable adapters is leveraged to extract features of the input contour sketch, reducing the number of trainable parameters. To maintain the geometric modelling features, edge-constraint initialization is introduced to optimize the strokes of guidance sketches.
3. Based on the objective and subjective results, the proposed method generates high-quality freehand sketches with more details, achieving better generative performance than existing methods.
4. The paper is generally well-written.

**Limitations:**

1. Although this work explores an interesting task (the generation of freehand sketches of mechanical components), it only involves a single modality.
2. The proposed model structure is not novel because most components have been widely used before such as the vision transformer for feature extraction and the transformer decoder for the generator.
3. The motivation behind the model design is not very clear. Have the diffusion-based methods been explored or compared?

**Suitability:**

1

---

### Meta-Review · Area_Chair_ar2m · 2024-06-27

**Recommendation:** Accept (Poster)
**Confidence:** 4

**Metareview:**

This paper was reviewed by four experts in the field. The paper received mixed reviews WR, BA, WA, WA.
The reviewers liked the idea of this paper, thorough evaluation and good presenation.
The reviewers also raised the following concerns on the application scenarios, comparisons with other baselines.
Based on the rebuttal, the AC feels that the added experiments on recognition, retrieval and more comparisons with Clipascene, LBS +vectorization well solve the concerns. Based on the reviews, the AC would like to recommend the acceptance of this paper, and suggest the authors to include the added experiments in the rebuttal to the final version.

---

### Meta-Review · Senior_Area_Chairs · 2024-07-10

**Recommendation:** Accept (Poster)
**Confidence:** 4

**Metareview:**

This paper received mixed ratings initially. After rebuttal, most reviewers are satisfied with the response and tend to accept the paper, while one reviewer still questioned application isse. SAC and AC carefully checked the reviews and rebuttal and recommend acceptance of the paper.